# Estimates of the basic reproduction number for rubella using seroprevalence data and indicator-based approaches

**Timos Papadopoulos[1,2]\*, Emilia Vynnycky[1,3,4]**

**1** Modelling and Economics Unit, UK Health Security Agency, London, United Kingdom, **2** Institute of Sound and Vibration Research, University of Southampton, Southampton, United Kingdom, **3** TB Modelling Group and TB Centre, London School of Hygiene & Tropical Medicine, London, United Kingdom, **4** Centre for Mathematical Modelling of Infectious Diseases, Faculty of Epidemiology and Population Health, London School of Hygiene & Tropical Medicine, London, United Kingdom

\* Timos.papadopoulos@phe.gov.uk

## Abstract

The basic reproduction number ($R_0$) of an infection determines the impact of its control. For many endemic infections, $R_0$ is often estimated from appropriate country-specific seroprevalence data. Studies sometimes pool estimates from the same region for settings lacking seroprevalence data, but the reliability of this approach is unclear. Plausibly, indicator-based approaches could predict $R_0$ for such settings. We calculated $R_0$ for rubella for 98 settings and correlated its value against 66 demographic, economic, education, housing and health-related indicators. We also trained a random forest regression algorithm using these indicators as the input and $R_0$ as the output. We used the mean-square error to compare the performances of the random forest, simple linear regression and a regional averaging method in predicting $R_0$ using 4-fold cross validation. $R_0$ was <5, 5–10 and >10 for 81, 14 and 3 settings respectively, with no apparent regional differences and in the limited available data, it was usually lower for rural than urban areas. $R_0$ was most correlated with educational attainment, and household indicators for the Pearson and Spearman correlation coefficients respectively and with poverty-related indicators followed by the crude death rate considering the Maximum Information Coefficient, although the correlation for each was relatively weak (Pearson correlation coefficient: 0.4, 95%CI: (0.24,0.48) for educational attainment). A random forest did not perform better in predicting $R_0$ than simple linear regression, depending on the subsets of training indicators and studies, and neither out-performed a regional averaging approach. $R_0$ for rubella is typically low and using indicators to estimate its value is not straightforward. A regional averaging approach may provide as reliable an estimate of $R_0$ for settings lacking seroprevalence data as one based on indicators. The findings may be relevant for other infections and studies estimating the disease burden and the impact of interventions for settings lacking seroprevalence data.

**Data Availability Statement:** In the 'Methods - Data sources' section of the manuscript and specifically in the sections 'Seroprevalence data',

'Demography' and 'Indicators' we give complete references with URL links from where all the data that were used in the paper are freely available to download. The data and code that we used to create our results are available at https://github.com/timosp/Estimates-basic-reproduction-number-rubella.git.

**Funding:** This work was supported by funding from GAVI, the Vaccine Alliance, via the Vaccine Impact Modelling Consortium (VIMC, www.vaccineimpact.org). VIMC is jointly funded by Gavi, the Vaccine Alliance, and by the Bill Melinda Gates Foundation (BMGF grant number: OPP1157270). This work was carried out as part of the Vaccine Impact Modelling Consortium (www.vaccineimpact.org), but the views expressed are those of the authors and not necessarily those of the Consortium or its funders. The funders were given the opportunity to review this paper prior to publication, but the final decision on the content of the publication was taken by the authors. The funders had no role in study design, data collection and analysis, decision to publish, or preparation of the manuscript.

**Competing interests:** The authors have declared that no competing interests exist.

## Author summary

The basic reproduction number ($R_0$) of an infection, defined as the average number of secondary infectious people resulting from the introduction of an infectious person into a totally susceptible, determines how easily the infection can be controlled. For many established endemic infections, $R_0$ is estimated using data describing the presence of antibodies in a population obtained prior to the introduction of vaccination in that population (pre-vaccination seroprevalence data). For countries lacking such data the estimation is often done by pooling estimates from their geographical region. We estimated $R_0$ for rubella for 98 settings with existing prevaccination seroprevalence data and we investigated the effectiveness of using simple machine learning regression methods to predict $R_0$ from 66 demographic, economic, education, housing and health-related indicators in those same settings. Our results suggest that the indicator data and prediction methods under investigation do not perform better than regional pooling. We discuss possible ways of improving the prediction accuracy. Since research on predicting $R_0$ using socio-economic data is very scarce, our findings may also be relevant to estimating the disease burden and the impact of interventions in other pathogens.

## Introduction

The impact of an intervention against an infection greatly depends on the basic reproduction number ($R_0$) of the associated pathogen, defined as the average number of secondary infectious people resulting from the introduction of a typical infectious person into a totally susceptible population[1]. For endemic vaccine-preventable infections, it is often estimated for a given setting from country-specific seroprevalence data collected before vaccination has been introduced[2], but it is unclear if estimates from one setting can be extrapolated to other countries. An understanding of how the basic reproduction number differs between settings is important for studies which try to either estimate the burden of an infectious disease or to predict the impact of interventions for settings without pre-existing seroprevalence data.

For settings without pre-vaccination seroprevalence data, studies sometimes base the pre-vaccination epidemiology and therefore $R_0$ on the regional average, calculated from the available data from all settings in the same geographical region[3–5]. It is also plausible that individual or combinations of socio-economic indicators could give insight into $R_0$ for a given setting. For example, the number of people that each person contacts might be expected to be correlated with factors such as the amount of crowding, which, in turn, is often correlated with other factors, such as poverty.

To our knowledge, the extent to which the regional average or particular indicators might predict what $R_0$ might be for a given setting has not been studied for any infection. The only related study is that of Santermans et al[6], which studied how differences in $R_0$ for varicella zoster for 9 European countries might be explained by differences in demographic, socio-economic and spatio-temporal indicators. The study found positive associations between $R_0$ and factors such as infant vaccination coverage for different vaccines and childcare attendance and negative associations with wealth inequality and poverty. As the study considered a small number of European countries and one infection, it is unclear whether the findings are generalisable either to other continents or to other infections.

In this study, we estimate the basic reproduction number for rubella for 98 settings from around the world and correlate its value against 66 demographic, economic, education, health and housing indicators. In addition, we compare the performance of simple linear regression

and random forest approaches in predicting $R_0$ from these 66 indicators against $R_0$ estimated using a regional averaging approach.

## Results

### Estimates of the basic reproduction number

Fig 1 summarizes estimates of the basic reproduction number for each study. The basic reproduction number was less than 5 for over half of the settings, with the point estimate being below 5 for 81 settings and the upper 95% confidence limit being below 5 for 62 settings respectively. The point estimate of the basic reproduction number was in the range 5–10 for 14 settings and exceeded 10 for just 3 settings, namely rural Chile, East Germany, and the Czech Republic before 1967, although the confidence intervals were very wide for each setting.

For most of the settings for which data were available for both urban and rural areas from the same year, the basic reproduction number was higher for urban than for rural areas, although the 95% confidence intervals sometimes overlapped. For urban and rural Peru before 1967, for example, it was 3.7 (95% CI: 3.3–4.4) and 2.2 (95% CI: 1.9–2.4) for respectively, and 6.5 (95% CI: 5.4–7.8) and 2.3 (95% CI 2.1–4.0) for urban and rural Uruguay before 1967 respectively. Exceptions to this pattern included Chile from before 1967 for which the basic reproduction number was 3.9 (95% CI: 3.5–5.2) and 16.5 (95% CI: 12.8–25.2) for urban and rural areas respectively.

In general, the value for the basic reproduction number that would have been estimated for each setting using the default method was approximately 2.3–3.3 for countries in the African, Eastern Mediterranean, Western Pacific and South East Asian regions, and around 2.7–3.8 and 3.7–6.7 for countries in the Americas and Europe respectively. The 95% range of the estimates based on the default approach, however, was very wide, with the upper limit of the 95% range reaching over 30 for several studies. The corresponding mean square error, calculated over all the studies, of $R_0$ for each setting based on the default approach compared to $R_0$ calculated using study-specific seroprevalence data was approximately 7, meaning that on average, estimates of $R_0$ of 4, for example, could be predicted to be as low as 1.5 or as high as 6.5. However, the error varied between regions (see S3 Table), with the lowest MSE of around 1 being predicted for the African and Western Pacific regions, intermediate values of 5 predicted for the Eastern Mediterranean and South East Asian regions, and high values of 9 and 23 predicted for the Americas and Europe. The range of the MSE was also very wide and differed between regions, with a 95% range of 1–16 and 13–695 from Africa and Europe respectively (S3 Table).

The same general patterns in the basic reproduction number were estimated for the alternative assumptions about contact between children and adults (S1 Fig), with $R_0$ being below 10 for most settings. For pessimistic assumptions about contact between children and adults, very high values for $R_0$ of above 20 were estimated for two settings, namely rural Chile and the Czech Republic before 1967. For both assumptions about contact, estimates of $R_0$ using the regional point estimate of the force of infection were usually smaller and led to a larger mean-square error than the default $R_0$ (Figs 1 and S1 and S3 Table).

### Correlation between $R_0$ calculated using study-specific seroprevalence data and indicators

In general, the indicators with which $R_0$, as estimated using study-specific seroprevalence data, was most correlated depended on the correlation coefficient and, when the correlation was calculated considering all 98 seroprevalence datasets, the correlation was generally weak

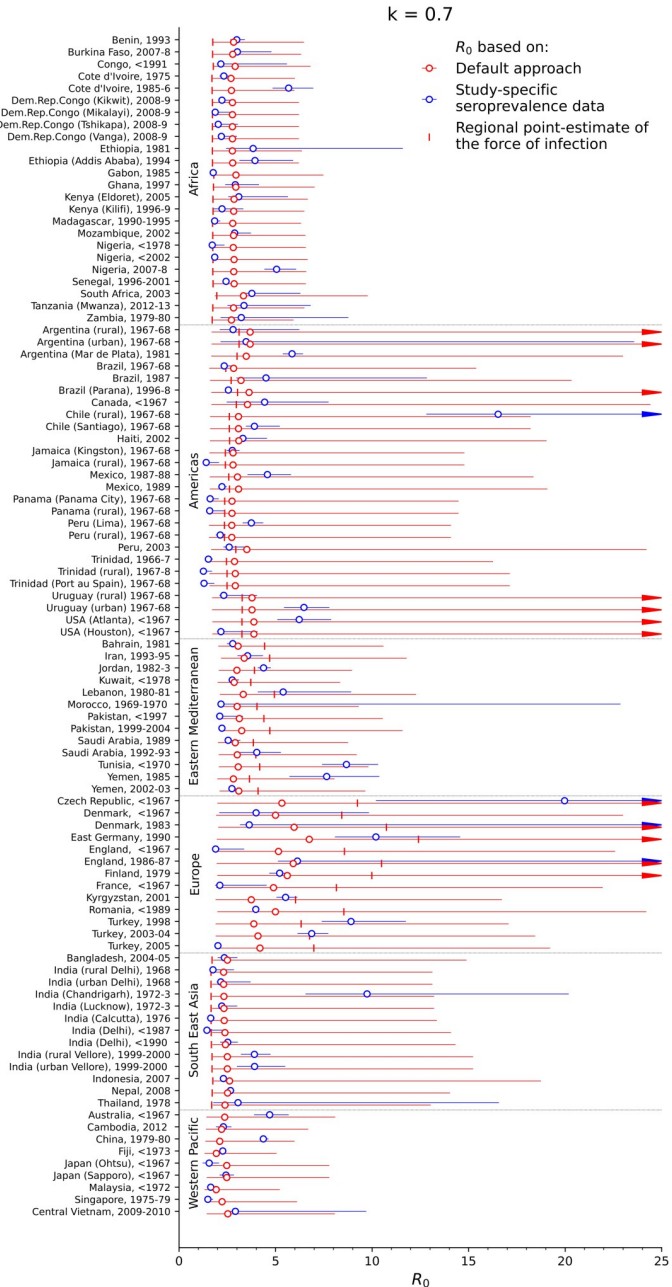

**Fig 1. Estimates of the basic reproduction number for each of the studies.** The blue circles reflect the values calculated from study-specific seroprevalence data (study-specific point estimates), the red circles reflect the default $R_0$ estimates and the vertical markers reflect $R_0$ estimated from the regional point estimate of the force of infection. The red and blue bars reflect the 95% ranges of the study-specific and default estimates based on bootstrapping.

(Tables 1 and S4, and Fig 2). For the Pearson correlation coefficient, the greatest correlation was obtained between the basic reproduction number and the educational attainment among people aged 25+ years (0.4 95% CI: 0.24, 0.48). For the Spearman correlation coefficient, three housing-related indicators had the highest correlation with the basic reproduction number, with the number of physicians per 1000 population having the next greatest correlation (0.32, 95% CI: 0.22,0.37). Considering the MIC, the basic reproduction number had the greatest

**Table 1. Summary of the top ten correlation coefficients and MIC for the association between the basic reproduction number and the indicators, obtained considering estimates of the basic reproduction number for all countries.** The columns labelled "CC" hold the coefficient, with the 95% range obtained by bootstrapping; the column labelled "N" holds the number of data points used to calculate the coefficient.

| Rank | Pearson | | | Spearman | | | MIC | | |
|---|---|---|---|---|---|---|---|---|---|
| | Indicator | CC | N | Indicator | CC | N | Indicator | CC | N |
| 1 | Educational attainment, at least completed upper secondary, population 25+, total (%) (cumulative) | 0.4 (0.24, 0.48) | 88 | Number of households 5 persons—Proportion over All households | -0.45 (-0.5, -0.36) | 56 | Poverty gap at $1.90 a day (2011 PPP) (%) | 0.37 (0.21, 0.34) | 92 |
| 2 | Educational attainment, at least completed upper secondary, population 25+, female (%) (cumulative) | 0.4 (0.25, 0.47) | 88 | Number of households 5 persons—Per capita | -0.41 (-0.46, -0.32) | 56 | Poverty headcount ratio at $5.50 a day (2011 PPP) (% of population) | 0.36 (0.22, 0.37) | 92 |
| 3 | Educational attainment, at least completed upper secondary, population 25+, male (%) (cumulative) | 0.39 (0.22, 0.47) | 88 | Number of households 6 persons and over—Per capita | -0.33 (-0.41, -0.22) | 53 | Crude death rate per 1000 population | 0.35 (0.2, 0.37) | 98 |
| 4 | Number of households 5 persons—Proportion over All households | -0.34 (-0.42, -0.19) | 56 | Physicians (per 1,000 people) | 0.32 (0.22, 0.37) | 98 | Life expectancy at birth (both sexes) | 0.34 (0.2, 0.33) | 98 |
| 5 | Physicians (per 1,000 people) | 0.33 (0.16, 0.41) | 98 | Prevalence of underweight, weight for age (% of children under 5) | -0.31 (-0.34, -0.21) | 94 | Poverty gap at $3.20 a day (2011 PPP) (% of population) | 0.34 (0.22, 0.36) | 92 |
| 6 | Proportion of the population aged 65+ | 0.32 (0.2, 0.51) | 98 | Immunization, measles (% of children ages 12–23 months) | 0.3 (0.22, 0.36) | 98 | Poverty headcount ratio at $1.90 a day (2011 PPP) (% of population) | 0.34 (0.22, 0.36) | 92 |
| 7 | Number of households 5 persons—Per capita | -0.28 (-0.35, -0.1) | 56 | Number of households 6 persons and over—Proportion over All households | -0.29 (-0.37, -0.18) | 53 | Poverty gap at $5.50 a day (2011 PPP) (% of population) | 0.33 (0.23, 0.37) | 92 |
| 8 | Proportion of the population aged 0–14 | -0.27 (-0.39, -0.14) | 98 | Health expenditure, total (% of GDP) | 0.28 (0.21, 0.35) | 98 | Poverty headcount ratio at $3.20 a day (2011 PPP) (% of population) | 0.33 (0.22, 0.37) | 92 |
| 9 | Population living in slums (% of urban population) | -0.27 (-0.32, -0.13) | 75 | Urban population (% of total) | 0.28 (0.2, 0.34) | 98 | Proportion of the population aged 0–4 | 0.32 (0.27, 0.38) | 98 |
| 10 | Number of households 6 persons and over—Proportion over All households | -0.26 (-0.34, -0.14) | 53 | Educational attainment, at least completed upper secondary, population 25+, total (%) (cumulative) | 0.27 (0.2, 0.35) | 88 | Physicians (per 1,000 people) | 0.31 (0.25, 0.37) | 98 |

correlation with two poverty-related indicators, namely Poverty gap at $1.90 a day (0.37, 95% CI: 0.21,0.34) and Poverty headcount ratio at $5.50 a day (0.36, 95% CI: 0.22,0.37).

When the simple correlation analyses were repeated by region, economic indicators were the most correlated with the basic reproduction number in Africa (Table 2). For the Americas, health-related indicators, for example, the percentage of children who were aged under 6 months who were breast-feeding, the percentage of the population that was urban and several poverty-related indicators were most correlated with the basic reproduction number (Table 2). For several indicators, the size of the correlation with $R_0$ was larger when they were calculated on a regional, compared to a global basis (Tables 1 and 2 and Fig 2). For the remaining regions, the small number of studies complicated the interpretation of the findings from linear regression (findings not shown).

## Comparison between $R_0$ predicted using simple linear regression and random forest regression

Figs 3 and 4 compare the performance of simple linear regression, random forest and the default method for calculating $R_0$, as quantified by the MSE over the 4-fold cross-validation

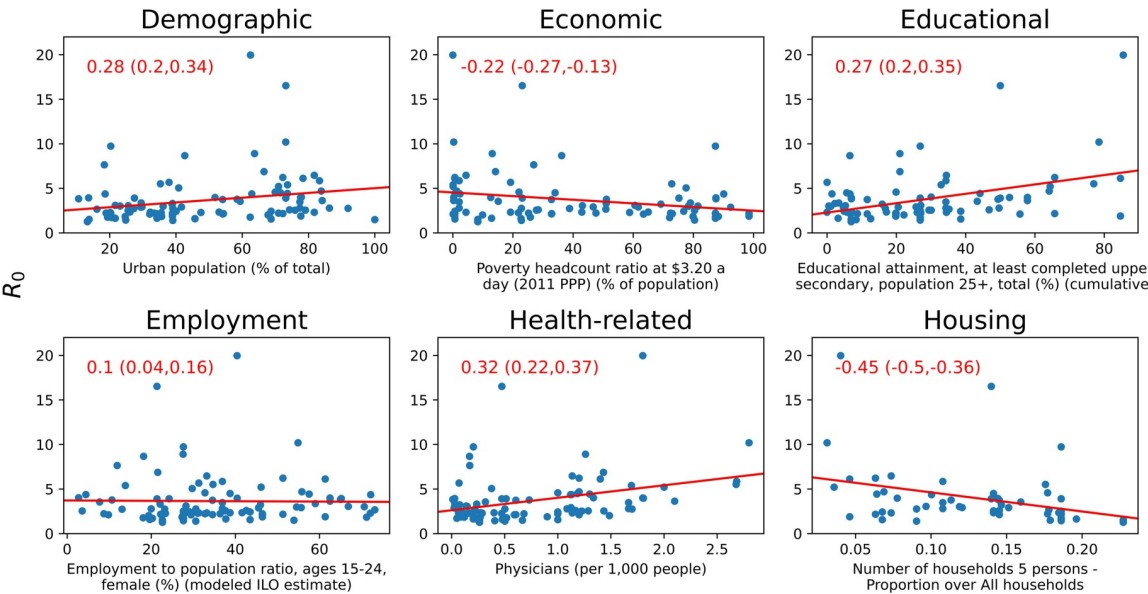

**Fig 2. Indicators in each category with the largest Spearman correlation with $R_0$ (correlation value and CI in red font).** The Pearson correlation regression line is plotted in red.

experiments, of the value predicted by these methods compared to that calculated using study-specific seroprevalence data. In general, the prediction error associated with using the default approach for calculating $R_0$ is comparable and in many instances slightly lower than the corresponding linear regression and random forest error, although the range of the error sometimes overlapped.

When the missing indicators were not imputed (Fig 3), the error associated with predictions based on linear regression differed between indicators. For most indicators, the MSE values range from approximately 7.5 to 9. The smallest error (MSE = 2.73) was associated with a poverty-related indicator ("Poverty gap at national poverty lines (%)"). The errors associated with four indicators related to schooling, undernourishment, breastfeeding and living in slums were also considerably lower than most, with a MSE of 5.35–5.5. The errors associated with the indicators for the number of doctors' consultations, number of people per room and all the indicators related to number of households with various numbers of persons are higher than all the rest, ranging in values of 11 and upwards. The min-max range of the error is fairly narrow and the minimum MSE in badly performing indicators is generally higher than the maximum MSE of better performing indicators.

In the random forest non-imputed results, the lowest average MSE equals 6.88 for the random forest trained and tested using the 63 indicators that have up to 70 missing values and the highest average MSE is equal to 13.8 for the random forest trained and tested using the 50 indicators with up to 40 missing values.

In contrast, in the imputed data results (plotted in Fig 4) the MSE for all linear regression, random forest and default methods is similar at around the value of 8.

As shown in Figs A and B in S2 Text much of the variation in the performance of the indicators in the simple linear regression and random forest analyses disappears once the two studies which have much higher values for $R_0$ than the other studies, namely, the Czech Republic and Chile (rural) ($R_0$ = 19.97 and $R_0$ = 16.53 respectively) are omitted from the experiments. The individual prediction error of those two studies drives the overall MSE of the 4-fold cross validation experiment upwards. As can be seen in Table A in S2 Text, the best performing

**Table 2. Summary of the top five correlation coefficients and MIC for the association between the basic reproduction number and the indicators, obtained considering estimates of the basic reproduction number just for countries in the African region or the Americas.** The columns labelled "CC" hold the coefficient, with the 95% range obtained by bootstrapping; the column labelled "N" holds the number of data points used to calculate the coefficient.

| Rank | Pearson | | | Spearman | | | MIC | | |
|---|---|---|---|---|---|---|---|---|---|
| | Indicator | CC | N | Indicator | CC | N | Indicator | CC | N |
| **African region** | | | | | | | | | |
| 1 | Unemployment, total (% of total labor force) (national estimate) | 0.35 (0.09, 0.48) | 24 | Poverty headcount ratio at national poverty lines (% of population) | -0.4 (-0.46, -0.18) | 24 | Number of households 3 persons—Proportion over All households | 1.0 (0.19, 1.0) | 6 |
| 2 | Poverty headcount ratio at national poverty lines (% of population) | -0.33 (-0.44, -0.08) | 24 | Income share held by highest 10% | 0.38 (0.17, 0.5) | 24 | Number of households 4 persons—Proportion over All households | 1.0 (0.19, 1.0) | 6 |
| 3 | Income share held by highest 10% | 0.31 (0.09, 0.54) | 24 | Health expenditure, total (% of GDP) | 0.38 (0.19, 0.52) | 24 | Immunization, measles (% of children ages 12–23 months) | 0.55 (0.23, 0.65) | 24 |
| 4 | Poverty gap at $1.90 a day (2011 PPP) (%) | -0.3 (-0.36, -0.08) | 24 | Income share held by highest 20% | 0.33 (0.15, 0.51) | 24 | Unemployment, total (% of total labor force) (national estimate) | 0.52 (0.28, 0.59) | 24 |
| 5 | Poverty gap at $3.20 a day (2011 PPP) (% of population) | -0.29 (-0.34, -0.06) | 24 | Educational attainment, at least completed upper secondary, population 25+, male (%) (cumulative) | -0.32 (-0.45, -0.06) | 19 | Physicians (per 1,000 people) | 0.49 (0.27, 0.57) | 24 |
| **Americas** | | | | | | | | | |
| 1 | Exclusive breastfeeding (% of children under 6 months) | 0.45 (0.3, 0.52) | 23 | Urban population (% of total) | 0.61 (0.53, 0.69) | 26 | Immunization, DPT (% of children ages 12–23 months) | 0.58 (0.36, 0.63) | 26 |
| 2 | Immunization, DPT (% of children ages 12–23 months) | 0.45 (0.19, 0.51) | 26 | Exclusive breastfeeding (% of children under 6 months) | 0.59 (0.52, 0.65) | 23 | Exclusive breastfeeding (% of children under 6 months) | 0.55 (0.42, 0.6) | 23 |
| 3 | Unemployment, total (% of total labor force) (modeled ILO estimate) | -0.42 (-0.5, -0.28) | 26 | Prevalence of underweight, weight for age (% of children under 5) | -0.57 (-0.67, -0.47) | 26 | Poverty gap at $3.20 a day (2011 PPP) (% of population) | 0.53 (0.41, 0.53) | 26 |
| 4 | Educational attainment, at least completed upper secondary, population 25+, total (%) (cumulative) | 0.42 (0.21, 0.51) | 24 | Unemployment, total (% of total labor force) (modeled ILO estimate) | -0.57 (-0.62, -0.48) | 26 | Poverty headcount ratio at $3.20 a day (2011 PPP) (% of population) | 0.53 (0.41, 0.53) | 26 |
| 5 | Urban population (% of total) | 0.42 (0.28, 0.51) | 26 | Immunization, DPT (% of children ages 12–23 months) | 0.49 (0.39, 0.59) | 26 | Adjusted net enrollment rate, primary (% of primary school age children) | 0.48 (0.32, 0.57) | 25 |

indicator is the only one having a missing value for both those studies. In addition, the difference in the MSE between the default approach for predicted $R_0$ and simple linear regression and random forest methods reduces once these two studies are omitted from the experiments. Further details of the effect of the presence or absence of these two studies in the folds are provided in S2 Text.

## Random forest parameter tuning

In general, optimising the set of parameters used for the random forest algorithm for predicting $R_0$ led to a small reduction in average MSE, from 9.87 for the default set of parameters to 8.86 in the optimised case (these values correspond to the average of the MSE for the 10 repetitions plotted in Fig 5). In addition, the latter of those values was still higher than that associated with the default approach used for calculating $R_0$, for which the MSE was 6.97 (as plotted in Fig 3). As shown in Fig 5, the optimal value for the MSE obtained with the nested-optimisation method for the random forest algorithm did not vary considerably across the 10 repetitions, although the minimum to maximum range of the MSE across the 96 different sets of parameters varied. A more detailed exposition of the nested cross-validation optimisation results is

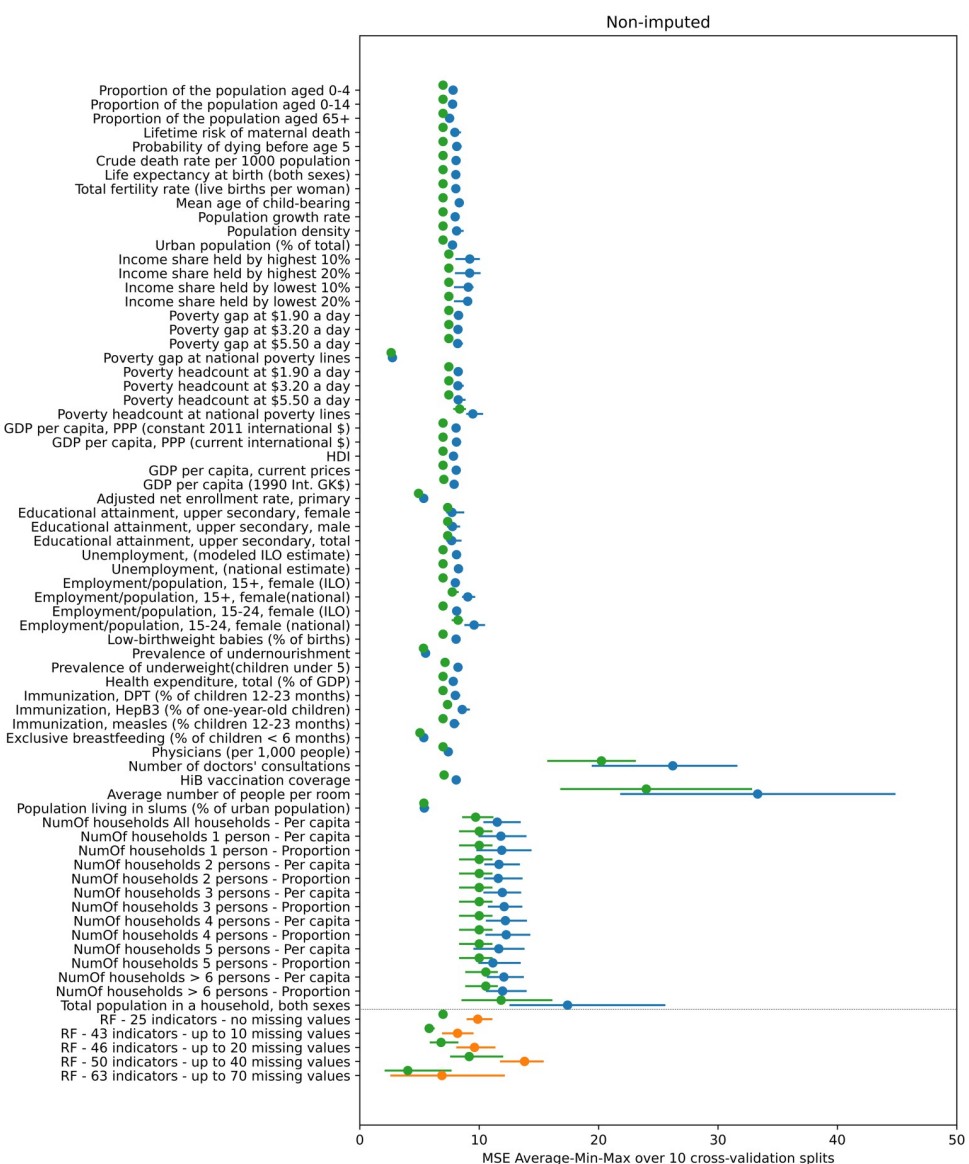

**Fig 3. Mean value (blue and orange dots) and minimum-maximum value range (blue and orange line) of the MSE of the predicted $R_0$ over the 10 repetitions of the 4-fold cross-validation experiment using the non-imputed dataset.** The blue color corresponds to the 66 linear regression results (one for each indicator) and the orange color to the 5 random forest results (one for each subset of indicators). The green dots indicate the average MSE for $R_0$ as calculated using the default approach compared to that calculated using study-specific seroprevalence data.

given in S3 Text. We note that the minimum end of the performance range (blue bars in Fig 5) is lower than the MSE error of the nested optimisation results in some of the 10 experiment repetitions. This is in accordance with previous theoretical and empirical comparison results between nested and non-nested optimisation methods[7].

## Discussion

Our analyses found that the basic reproduction number for rubella is typically low and under 5 for many settings, which is consistent with the view that rubella is less infectious than other childhood immunizing infections, such as measles and mumps. We also found that in many

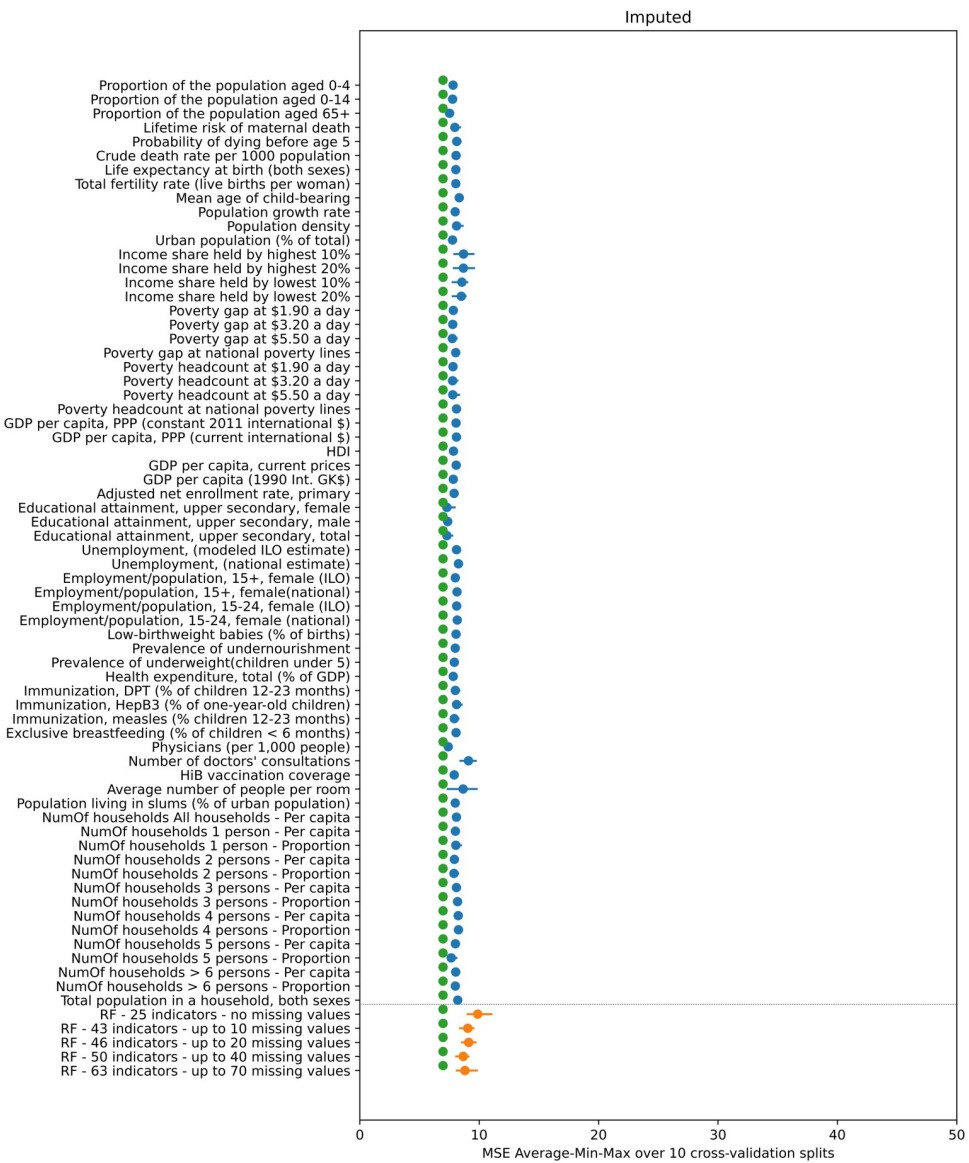

**Fig 4. Mean value (blue and orange dots) and minimum-maximum value range (blue and orange line) of the MSE of the predicted $R_0$ over the 10 repetitions of the 4-fold cross-validation experiment using the imputed dataset.** The blue color corresponds to the 66 linear regression results (one for each indicator) and the orange color to the 5 random forest results (one for each subset of indicators). The green dots indicate the average MSE for $R_0$ as calculated using the default approach compared to that calculated using study-specific seroprevalence data.

cases, the basic reproduction number was lower for rural than areas but that the correlation between $R_0$ and indicators was not straightforward. In general, the performance of a regional averaging approach to estimate the basic reproduction number was often better than that using linear regression and random forest approaches.

This is the first study to attempt to correlate the basic reproduction number for rubella in different time periods globally against different indicators. The only other related study to date, by Santermans et al considered varicella zoster and considered nine European settings [6]. The study found positive associations between $R_0$ and factors such as infant vaccination coverage for different vaccines and childcare attendance and negative associations with wealth

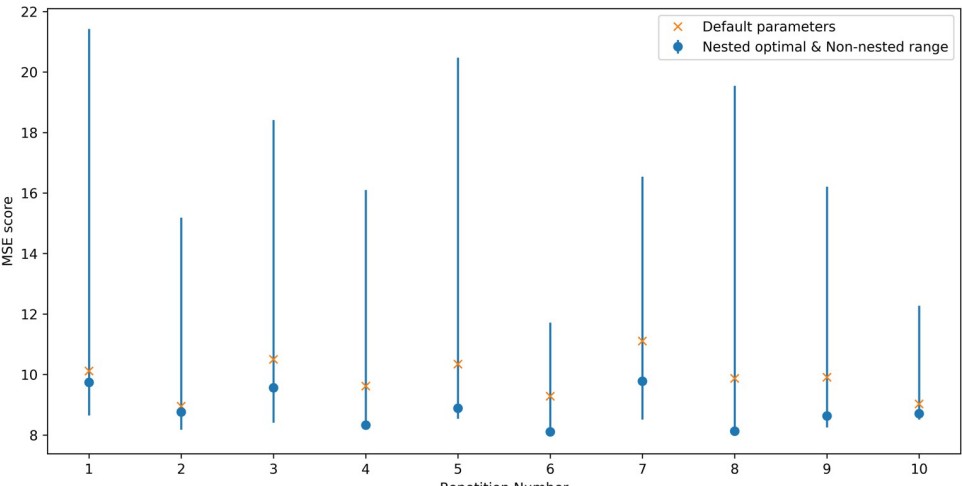

**Fig 5. Performance (as quantified by the MSE in each of the 10 splits to folds) of the default set of parameter values used in the random forest (yellow x) compared to the optimal parameters for the random forest identified through nested optimisation (blue circle).** Blue lines are the MSE minimum to maximum range for the 96 points of the parameters grid (hence this minimum is the optimal result of the non-nested optimization method).

inequality and poverty. As shown in our analyses, the correlation between indicators and $R_0$ for rubella is not straightforward, with the indicator with which $R_0$ had the greatest correlation depending both on the correlation statistic used and the region. In addition, the size of the correlation depended on the region considered, and was weaker when all study settings were considered than when the correlation was conducted on a regional basis. The size of the correlation in the latter case was comparable to that seen in Santermans et al[6].

The correlation between the basic reproduction number and several indicators was in the opposite direction from that expected. For example, in the simple linear regression analyses considering all countries, the basic reproduction number went down as the proportion of households which comprised 5 people increased and it increased as the number of physicians per 1000 population increased. This may have been due to other unknown factors confounding the observed relationship between $R_0$ and crowding indicators. It is also plausible that some outliers may have affected our results, but the extent to which this was the case is unclear. For example, considering Singapore, the estimated $R_0$ was low but the population density was much higher than in other settings.

Our finding that the basic reproduction number was typically higher in urban than in rural areas (Fig 1) is consistent with the fact that the people in urban areas are more likely to contact others than people in rural areas. One of the exceptions to this was Chile from before 1967, for which $R_0$ in rural areas was several times greater than that in rural areas. An unexpectedly high estimate in rural areas might occur if, for example, the seroprevalence data had been collected shortly after an epidemic had occurred, resulting in unusually high levels of seroprevalence. Small differences in the $R_0$ between urban and rural areas may also occur if rural residents often visit urban areas, because of work or other reasons.

To our knowledge, our analyses are the first to assess the performances of regional averaging (default), simple linear regression and random forest approaches in predicting $R_0$. We found that the default regional averaging approach employed in this work, whereby we calculated a set of $R_0$ estimates based on each force of infection estimate from the regional collection of bootstraps and then took the median of that $R_0$ estimate, performed better than the "regional point estimate of $R_0$". The under-performance of the latter is unsurprising, given its

method of computation using the regional "point estimates" of the force of infection. These were defined in previous analyses[5,8–10], in which the main outcome of interest of interest was the incidence of Congenital Rubella Syndrome (CRS), as the force of infection estimates which led to the median incidence of CRS per 100,000 live births in the absence of vaccination. Since the CRS incidence depends both on the proportion of women who reach child-bearing age still susceptible and the force of infection among adult women, the combination of force of infection estimates among children and adults, out of 1000 bootstrap-derived values, which results in the median CRS incidence is unlikely to result in the median estimate of the basic reproduction number.

Although the two machine learning approaches are technically more sophisticated than the default approach it is interesting that, when all studies were considered, they were often out-performed by the default approach, based on regional averaging. However, we also note that the two machine learning approaches were sensitive to inclusion of two datasets with particu-larly high $R_0$ values and once they were excluded, the performance of the three approaches was similar. These findings suggest that for studies of the disease burden or the impact of interven-tions against rubella, indictor-based approaches are unlikely to improve upon using a regional averaging approach for settings lacking seroprevalence data.

There are several limitations of our analyses, which could have contributed to the indica-tor-based approaches under-performing when predicting $R_0$. For example, for simplicity, the value for the indicator used when correlating the basic reproduction number was taken from the year in which the study was conducted. In practice, the seroprevalence in the study year results from the exposure to rubella infection over the lifetime of the study population. In addi-tion, when calculating the basic reproduction number, we just used one assumption for the contact between adults and children. Whilst that assumption is based on estimates from social contact studies from the past decade, it is plausible that for some of the settings in our study, such as those from many years ago, the amount of contact between people would have been different compared to that in recent years. In addition, due to the limitations of the seropreva-lence data, which typically comprised convenience samples covering three or four age groups, we used a crude age stratification when calculating $R_0$, just considering two broad age groups.

Our estimates of $R_0$ using the default approach involved taking the median of $R_0$ values as calculated using bootstrap-derived estimates of the force of infection from all studies in the same World Health Organization region. The performance of this approach may have been overestimated given that bootstrap-derived estimates used to calculate the median would have included those from each of the studies being used to evaluate the performance. In practice, the overestimate is likely to have been relatively small, given that at least 10 studies would have contributed to the bootstrap samples in each region. In order to corroborate this, we included in our analyses a calculation of the default $R_0$ estimates for all the studies of one region (EMRO region) in which the force of infection bootstraps from each study are excluded from the col-lection of bootstraps used to derive the default $R_0$ estimate of that particular study (see S4 Text). As is seen in Fig A in S4 Text, the impact of this exclusion is very small both in the cases where the default $R_0$ estimate is close and further way from study-specific $R_0$ estimate value. In addition to that, we note that in analyses of the global burden of CRS, which included the data-sets using the ones in this study, excluding a single seroprevalence dataset had little impact on estimates[4].

In a related manner, the comparison of the performance of the default approach to those of the Machine Learning methods using a cross-validation methodology is possibly giving an advantage to the former. This is because the default $R_0$ estimates for each study are calculated once, prior to the cross-validation split to folds, and they make use of information in studies from the same region regardless of whether these fall in the test or training set in each

particular split to folds. A more accurate version of that comparison would be to modify the default approach so that for each different indicator and each different split to cross-validation folds, a different default $R_0$ estimate is calculated for each study by excluding from the regional collection of force of infection bootstraps the studies that are in the same test fold (this exclusion would be dependent on the different indicators because of the missing values issue). However, given the minimal impact of excluding force of infection estimates of one study at a time (see discussion above and in S4 Text), and the increased complexity of the aforementioned modification, we did not pursue it.

The method we used to calculate the study-specific $R_0$ estimates had a few limitations: Our analyses did not take into account the genetic diversity of the Rubella virus[11] which may be a source of variability of the basic reproduction number over different settings, that could not have been predicted by the indicator that we used as inputs to the machine learning methods we examined. Another limitation is that, while the seroprevalence data used to calculate the force of infection estimates used in the $R_0$ typically spanned three or more age groups we used two age groups to describe the age-mixing patterns. The influence of more complex mixing patterns has been described elsewhere[12]. Finally, the possible effect of different assays in the seroprevalence data that we used for the determination of the study-specific $R_0$ estimates[13] may have introduced a source of variability not predictable by our machine learning methods.

Several further refinements of both the default and indicator-based approaches for predicting $R_0$ are potentially feasible, although the effect on their performance is unclear. First, the default approach could be refined to use alternative methods for grouping the countries, such as those developed for Global Burden of Disease analyses, which are based on both epidemiological aspects and geographical location. However, application of this method for grouping countries is complicated by the fact that some groupings include only a few countries each of which may have no pre-vaccination seroprevalence data.

Second, the parameter optimisation experiment that we ran on the random forest regression algorithm was limited to only 6 parameters and to 2 or 3 values for each of those parameters. A search over a more extensive parameter grid (even though much more computationally intensive) could result in a reduction of the expected prediction error of the random forest algorithm. However, the small scale parameter optimisation conducted here suggests that the prediction results obtained with the random forest prediction method are close to the optimum and that not much further improvement can be expected by a finer or wider parameter grid search.

Third, the effect of different methods for imputing missing values could be explored. As presented in the results, treating missing values with median value imputation has a significant effect on the prediction performance (Figs 3 and 4) and the effect of different methods of imputation on the prediction performance could be investigated.

Fourth, the prediction methods that we considered are unable to incorporate the uncertainty inherent in our data, which is present both in the indicator data that we use as input and in the estimated $R_0$ values that we use as the ground truth. Considering input data, the uncertainty can be quantified as the number of years of difference between the seroprevalence and the indicator data and this could also be modelled to fold in the case of missing data. In the case of the output the uncertainty can be quantified by the bootstrap-derived data distribution for different values of the contact parameter $k$. A Gaussian process methodology could be employed to incorporate those measures of uncertainty in the regression problem. The main advantage of such a method would be that it can provide a measure of uncertainty of the predicted results. For the prediction to be improved, it would be important for these measures of uncertainty to overcome one of the limitations of each of the prediction methods, namely that

they are unable to correctly predict the very high values of $R_0$ for some settings (particularly for those in which $R_0 > 10$).

In conclusion, our analyses confirm the view that the basic reproduction number for rubella is low and that an approach based on regional averaging for settings lacking prevaccination seroprevalence is likely to perform as well as, if not better, than indicator-based approaches involving simple linear regression or random forest regression algorithms. It is unclear whether refinement of these methods could lead to an improvement in their performance. Whilst our analyses have focused on rubella, these findings are likely to be relevant for studies of the disease burden or the impact of interventions for other infections.

## Methods

### Data sources

**Seroprevalence data.**    The analyses use force of infection estimates (defined as the rate at which susceptible people are infected) for 98 rubella seroprevalence datasets that were identified in a systematic review and had been used previously in estimates of the global burden of Congenital Rubella Syndrome (CRS) and the impact of measles-rubella vaccination[4,5]. The seroprevalence datasets are listed in Table A in S1 Table and were collected from 24, 26, 13, 13, 13 and 9 settings from the African, the Americas, Eastern Mediterranean, Europe, South East Asia and Western Pacific World Health Organization regions respectively before the introduction of rubella vaccination.

**Demography.**    The age distribution considering single year age bands of each of the countries used in calculating the basic reproduction number was based on UN population data[14] for the year in which the study was conducted or the publication year, if the study year was unavailable. For instances in which the number of people aged 80 years and over is not stratified into single year age bands, we calculated this stratification using UN population data on the number of survivors by age.

**Indicators.**    For each of the 98 seroprevalence studies, we extracted 66 demographic, economic, education, health and housing indicators for the year in which the study was conducted, where possible. The indicators expand on 39 indicators used by Santermans et al in their study of $R_0$ for varicella-zoster[6] and are summarized in Table A in S2 Table. For studies for which the study year was not known, the indicator was extracted for the year in which the data were published. If the value of the indicator for the actual study year was not available, the indicator was extracted for the year closest to the study year. Educational and employment were extracted from World Bank sources[15]. Economic and health-related indicators were extracted mainly from World Bank sources[15] but also from United Nations, International Monetary Fund, Organisation for Economic Co-operation and Development, World Health Organization and other sources[16–20]. Demographic data were extracted from United Nations population data and World Bank sources[14,15]. Housing data were extracted from World Bank and United Nations Statistics Division sources[15,21,22]. Table A in S2 Table lists the data source for each indicator and it summarizes the completeness of data on the indicators for the studies and the number of studies for which values for the indicators in question were available within 5 years, 5–10 and >10 years of the study year. The number of indicator values available for each study are provided in Table B in S2 Table. Values for the indicators were available for almost all of the studies, except for many of the housing-related indicators, which were not available for approximately 45% of the study settings. Values for all but one of the demographic indicators were available within 5 years of the study being conducted for all settings, and values for some of the economic indicators were only available more than 10 years after the study was conducted for 50% of the studies.

## Estimating the basic reproduction number

For each of the 98 seroprevalence studies, point estimates of the force of infection among those aged $<13$ and $\geq 13$ years, as published elsewhere[4,5], were used to calculate the basic reproduction number, as the dominant eigenvalue of the Next Generation Matrix[1]. We refer to the estimates of $R_0$ derived from the point estimates of the force of infection from each study as "study-specific point estimates of $R_0$". Additionally, 1000 bootstrap-derived samples for the force of infection (calculated as described in[4]) were used to derive 1000 bootstrap-derived estimates of the study-specific $R_0$ and, subsequently, the 95% range of the study-specific $R_0$ for each study. The forces of infection for each study are summarised in Fig 6. The matrix of Who Acquires Infection from Whom was assumed to have the following structure:

$$\begin{pmatrix} \beta_1 & k\beta_2 \\ k\beta_2 & \beta_2 \end{pmatrix}$$

The effective contact rate differs between $<13$ and $\geq 13$ year olds, with its relative size between children and adults, compared to that between adults (k), assumed to equal 0.7, based on contact survey data[23]. In sensitivity analyses, we explored the effect of using pessimistic assumptions about mixing between younger and older people on estimates of the basic reproduction number, using a value of 0.3 for k. The age distribution of the population used in calculating the basic reproduction number was based on UN population data[14] for the year in which the study was conducted or the publication year, if the study year was unavailable.

Previous studies have approximated the epidemiology of rubella in settings lacking seroprevalence by using a regionally averaging approach, using 1000 bootstrap-derived samples of the forces of infection compiled from all the studies from the same World Health Organization region[4,5]. To evaluate this approach, for each of the 98 studies we calculated what the median $R_0$ and its 95% range would have been estimated to be from these 1000 bootstrap-derived force of infection samples drawn from all the countries of the corresponding region. The bootstraps drawn from each study to create the each of those regional collection of 1000

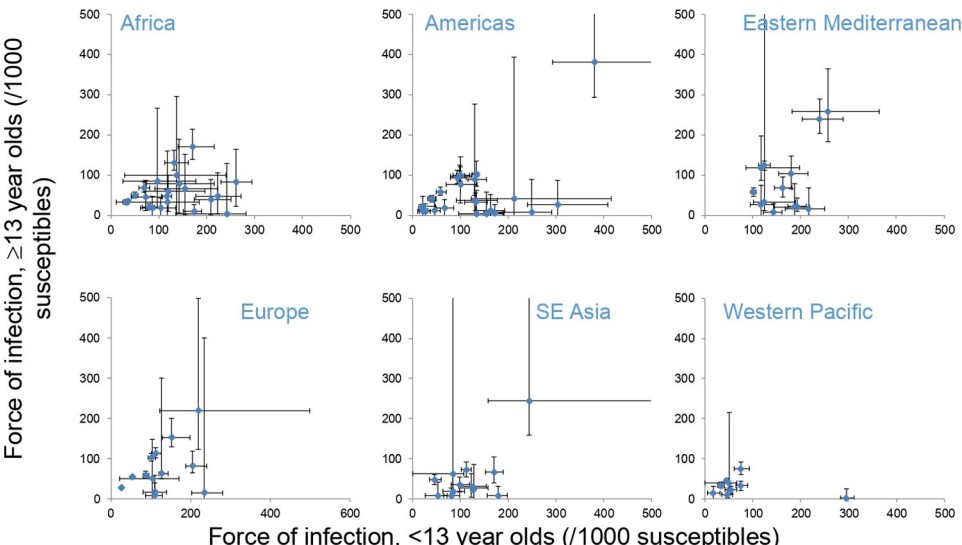

**Fig 6. Summary of the force of infection estimates used to calculate the basic reproduction number for each study in each region.** The bars reflect the 95% ranges based on bootstrapping.

force of infection bootstrap values is described in detail in references[4,5]. We refer to this median $R_0$ as the "default $R_0$" throughout the text. These default estimates of $R_0$ for the 98 studies are the same throughout our analyses including the cross-validation results we present. The error associated with the default $R_0$ and that calculated using the study-specific point estimate of $R_0$ was quantified as the mean square error (MSE) over the 98 studies. The 95% range of the MSE was calculated using 1000 bootstrap-derived estimates of the MSE computed from the corresponding $R_0$ estimates that were computed from 1000 bootstrap-derived force of infection.

In sensitivity analyses, we also calculate what $R_0$ for a given setting would have been estimated to be using a point estimate of the age-dependent force for the region, which has previously been used to obtain an approximate estimate of the average CRS incidence for countries lacking seroprevalence data[5,8]. This regional point estimate of the force of infection was taken as the value which led to the median burden of CRS out of the 1000 bootstraps in the absence of vaccination (see supplement in ref[5]). We refer to the estimate of the basic reproduction number obtained using the regional point estimate of the force of infection as the "regional point estimate of $R_0$"

## Measures of relation between indicators and $R_0$ estimates

The Pearson and Spearman correlation coefficients and the Maximum Information Coefficient (MIC) were calculated for the correlation between the basic reproduction number for each study and each of the 66 demographic, economic, education, health and housing indicators extracted for each study. The MIC has been used elsewhere in similar analyses for varicella zoster[6] and quantifies the correlation between two statistics when they may be non-linearly related[24]. The 95% range of the correlation coefficients were calculated using the 1000 bootstrap-derived values for the basic reproduction number. In the first instance, the correlation coefficients were calculated considering the basic reproduction numbers for all countries. In sensitivity analyses, the regression was repeated on a regional basis. In the results we present which rank the level of relation between the different indicators and $R_0$ from highest to lowest, we rank lowest all the indicators for which the 95% CI overlaps zero (i.e. the 2.5% and 97.5% percentiles of the bootstrap values are of opposite sign).

## Out-of-sample prediction

**Overview.**   We explored whether the basic reproduction number for a given setting could be predicted using either simple linear regression on any of the 66 indicators or a Random Forest regression algorithm trained on different subsets of the indicators. We used 4-fold cross-validation[25] to score both approaches and explored the effect of imputation. Further details of the cross-validation and imputation are provided below.

**Four-fold cross-validation.**   The 4-fold cross-validation process was similar for both the simple linear regression and random forest algorithms. A detailed description of the process ("experiment") is provided in S1 Text and we summarize it briefly here.

First, the values for $R_0$ that had been estimated using study-specific seroprevalence data for the 98 studies, along with their indicators, were split into 4 partitions ("folds"). Next, for the case of simple linear regression and for a given indicator, the values in three of the folds were used as the training dataset to establish a relationship between the point estimate of $R_0$ that had been estimated using seroprevalence data, and that indicator. The values in the fourth fold were then used as the test dataset, whereby the value of the indicator for each study was used to predict what $R_0$ might be expected to be in that study using the linear regression model that had been estimated using the training dataset. We then calculated the mean square error

(MSE) for the test dataset in these predicted $R_0$ values compared to those estimated using sero-prevalence data. These steps were repeated using each fold in turn as the test dataset and the remaining three as the train dataset, and we computed the average MSE for the four folds. These steps were repeated for each of the indicators in turn.

The four-fold cross-validation was repeated using 10 randomly generated splits of the 98 $R_0$ estimates to 4 folds, resulting in one value of the average MSE for each split and in the final stage of this process, we computed the average, minimum and maximum of those 10 average MSE values of the predicted $R_0$ compared to that calculated from study-specific seroprevalence data. We discarded splits to folds that had insufficient numbers of indicator values in each fold to train and test the linear regression algorithms (see further details below).

We used the steps and the same splits to folds as described above for estimating the performance of the random forest regression algorithm, except that subsets of indicators (rather than a single indicator at a time) are used train and test the random forest regression algorithm, as described below.

**Dealing with missing values in the indicator data.** Most of the 98 studies lacked values for some of the indicators (see Table B in S2 Table) and we explored the effect of both not imputing the missing values and of imputing them as follows.

A. *Non-imputation approach*. Not imputing the missing values has implications in the 4-fold cross-validation experiments. Considering the situation for simple linear regression, we note that the situation may arise where for a given indicator and a given split of the 98 studies to 4 folds, the training dataset has valid values for only one or none of the studies in it. In such cases it is not possible to fit a regression line. To address this, we generated a series of splits to folds and we kept the first 10 splits for which each indicator had at least one study with a valid value in each of the 4 folds (see S1 Text for further details). This guarantees that for all folds and all indicators there are at least 3 studies with valid values in the training dataset which is sufficient for fitting a regression line and that there is at least one study with a valid value in the test dataset. With this arrangement, a study that has a valid value for a given indicator is used exactly once for testing and three times for training the linear regression model corresponding to that indicator in each of the 10 repetitions of the 4-fold cross-validation experiment.

For consistency, we used the same 10 splits to folds for the random forest 4-fold cross-validation experiments. Since random forest regression algorithms use all available indicators to train and predict the outcome of interest, rather than just one, as is the case for simple linear regression, the treatment of the missing data was different from that used for simple linear regression. Instead, we considered the following 5 different subsets of indicators and we ran a separate cross-validation experiment for each subset:

- The 25 indicators that have no missing values at all, i.e. all 98 studies have values for those indicators.

- The 43 indicators that have up to 10 missing values. There are 69 studies which have values for each of those 43 indicators.

- The 46 indicators that have up to 20 missing values. There are 52 studies which have values for each of those 46 indicators.

- The 50 indicators that have up to 40 missing values. There are 32 studies which have values for each of those 50 indicators.

- The 63 indicators that have up to 70 missing values. There are 20 studies which have values for each of those 63 indicators.

We list the number of studies in the training and testing subsets for all 10 splits to 4 folds for both the linear regression and the random forest models in Table A in S1 Text.

B. *Imputation approach*. In the second approach we use imputation and we replaced all missing values (if any) of each indicator with the median value of the existing values among the 98 studies for that particular indicator. The imputation is applied once to the whole dataset prior to splitting to folds and all other parts of the cross-validation. With this approach there are no missing values in the imputed dataset and we were able to conduct all training and testing parts of the 4-fold cross-validation experiments as usual. For consistency purposes we used the same choice of 10 4-fold splits of the non-imputed experiments that is described above.

**Comparison of the linear regression, random forest $R_0$ predictions against those from the default method.**   To assess whether the linear regression and random forest approaches performed better in predicting $R_0$ across the settings than using the default $R_0$ estimate, we computed the average, minimum and maximum values of MSE over the same test folds of the 10 cross-validation splits described above for the default $R_0$ estimate compared to $R_0$ estimated using study-specific seroprevalence data. The resulting MSE values were compared against those calculated considering the $R_0$ for the linear regression and random forest approaches. As described above, $R_0$ calculated using the default approach is taken as the median $R_0$ calculated from 1000 bootstrap-derived samples of the forces of infection compiled from all the studies from the same World Health Organization region.

## Random forest optimisation

Unlike the fitting of a linear regression model, which is solely a function of the training dataset, the process of training a random forest prediction model is also a function of the choice of hyper-parameters of the random forest. For the prediction performance comparison results that we described above we used the default parameter values of Python's scikit-learn toolbox. Further to that, we also present results of a separate small-scale parameter optimisation run. For that parameter optimisation experiment we used the same 10 cross-validation splits to 4 folds described above and the first indicators' subset listed above, i.e. the one comprising the 25 indicators that have no missing values. We present non-nested optimisation results as well as nested cross-validation optimisation results (see references[7,26]). The former method amounts to simply selecting the parameter setup which achieves the lowest average MSE for a given split to folds. That method effectively violates the out-of-sample principle and can give over-optimistic estimates of the expected prediction performance in previously unseen test data. The latter case amounts to the results obtained when a separate inner-loop cross-validation experiment is run for each cross-validation fold and it is considered a more valid method to estimate the generalisation of the prediction performance in previously unseen test data.

## Software

All computations were carried out in a reproducible manner using the corresponding methods and functions of Python's scikit-learn package[26]. The 10 splits to folds were obtained by use of the 'sklearn.model_selection.KFold' class by setting the 'random_state' parameter which corresponds to the seed of the pseudorandom generator to increasing integer values starting from 0. The values corresponding to folds that were used in the results are given in Table A in S1 Text. The linear regression and random forest prediction results were obtained with the

methods of the 'sklearn.linear_model.LinearRegression' and 'sklearn.ensemble.RandomForestRegressor' classes. The missing values imputation described in the 'Imputation approach' section was implemented using the 'sklearn.impute.SimpleImputer' class. For the random forest parameter tuning results we used a 6-parameter grid comprising 96 parameters' choices (details are given in S3 Text).

We note that due to the missing values in the dataset we could not use the higher-level methods of the scikit-learn package (e.g. 'sklearn.model_selection.cross_val_score') and that for all the results we had to write code that implements the complete cross-validation methodology.

## Supporting information

**S1 Fig. Estimates of $R_0$ for pessimistic assumptions about contact. Fig A:** Estimates of the basic reproduction number for rubella, as calculated using seroprevalence data (blue circles) and the default approach (red circles) for each study for pessimistic assumptions about the amount of contact between children and adults $k = 0.3$). The vertical marks indicate $R_0$ as calculated using the regional point estimate for the force of infection.
(PDF)

**S1 Table. Summary of the study settings. Table A**: Summary of the sources of seroprevalence data from each region for the force of infection estimates for each setting, that is used in this analysis. Force of infection estimates from all of these datasets have been previously used in estimating the global burden of Congenital Rubella Syndrome or in the impact of Measles-Rubella vaccination.
(PDF)

**S2 Table. Completeness of the indicator data. Table A**: Number of studies for which the year in which the indicator was available differed by less than 5 years, 5–10 years or more than 10 years from the year in which the data were collected. For studies with an unspecified year of data collection, the difference was calculated as the difference between the publication year and the year for which the indicator was extracted. The right-most column cites the source from which the data was obtained. **Table B**: Number of missing indicators for each of the 98 studies.
(PDF)

**S3 Table. Estimates of the mean square error associated with $R_0$ estimates. Table A:** Estimates of the mean square error associated with $R_0$ estimates. Column 2 shows the mean square error of $R_0$, as calculated using the default approach against $R_0$ calculated using study-specific seroprevalence data. Columns 2–7 hold the minimum, median, maximum 2.5th and 95.5th percentiles of the MSE values between the 1000 bootstrap-derived values of $R_0$ calculated using the regional collection of force of infection bootstrap values and the 1000 bootstrap-derived study-specific $R_0$ values. The final column holds the MSE between study-specific $R_0$ and the regional point estimate of $R_0$.
(PDF)

**S4 Table. Correlation between $R_0$ and the indicators.** Table A: Summary of the correlation coefficients and MIC for the association between the basic reproduction number and the indicators. ranked in decreasing order. The columns labelled "CC" hold the coefficient, with the 95% range obtained by bootstrapping; the column labelled "N" holds the number of data points used to calculate the coefficient.
(PDF)

**S1 Text. Details of 4-fold cross-validation–Dealing with missing indicators in simple linear regression and random forest analyses. Fig A:** Examples of two different splits to folds for the toy-case cross-validation scenario described above. The 'x' marks denote missing values. **Table A:** Number of studies with a valid indicator value in each fold for each of the 10 repetitions of the 4-fold cross validation experiment. The indicators order for the Simple Linear Regression case is the same as in Table A in S2 Table and the order of indicators subsets for the Random Forest case is the same as in the main text. The row labelled 'Seed' gives the seed numbers used for each of the 10 split to folds (omitted seed numbers corresponds to splits that did not have at least one study with a valid (non-missing) value in each fold for each one of the 66 indicators.
(PDF)

**S2 Text. Effect of studies with high $R_0$ values on the performance of simple linear regression and Random Forest prediction. Table A:** Presence or absence of the two studies with highest value of $R_0$ from non-imputed linear regression 4-fold cross validation experiments. **Table B:** Presence or absence of the two studies with highest value of $R_0$ from non-imputed random forest 4-fold cross validation experiments. **Fig A**: Similar to Fig 3 of the main text but with the two studies ('Czech Republic, <1967' and 'Chile (rural), 1967–68') excluded. Mean value (blue and orange dots) and minimum-maximum value range (blue and orange line) of the MSE of the predicted $R_0$ over ten 4-fold cross validation splits. **Fig B:** Similar to Fig 4 of the main text but with the two studies ('Czech Republic, <1967' and 'Chile (rural), 1967–68') excluded. Mean value (blue and orange dots) and minimum-maximum value range (blue and orange line) of the MSE of the predicted $R_0$ over ten 4-fold cross validation splits.
(PDF)

**S3 Text. Random forest parameter tuning. Table A**: MSE performance and rank of the 96 parameter sets in the double-loop nested cross-validation experiment. The default parameter setup is in bold font.
(PDF)

**S4 Text.** Including or excluding the force of infection bootstraps of a study in the calculation of its default $R_0$ estimate **Fig A:** Estimates of the basic reproduction number for each of the studies in the EMRO region. The blue and red circles and bars are identical as in Fig 1 of the main text. The black circles and ranges are the default $R_0$ estimates and 95% CI that we obtain for each study when we exclude bootstrap values from that study from the collection of 1000 force of infection bootstrap used in the calculation. **Table A**: Comparison of default $R_0$ values with the values obtained when the regional collection of 1000 force of infections bootstrap values does not include those coming from the study for which the default $R_0$ is calculated.
(PDF)

## Author Contributions

**Conceptualization:** Timos Papadopoulos, Emilia Vynnycky.

**Data curation:** Timos Papadopoulos.

**Funding acquisition:** Emilia Vynnycky.

**Investigation:** Timos Papadopoulos, Emilia Vynnycky.

**Methodology:** Timos Papadopoulos, Emilia Vynnycky.

**Project administration:** Emilia Vynnycky.

**Software:** Timos Papadopoulos.

**Visualization:** Timos Papadopoulos.

**Writing – original draft:** Timos Papadopoulos, Emilia Vynnycky.

**Writing – review & editing:** Timos Papadopoulos, Emilia Vynnycky.

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
