## [Decision Letter · Decision Letter 0]

18 Jul 2021

Dear Dr Papadopoulos,

Thank you very much for submitting your manuscript "Estimates of the basic reproduction number for rubella using seroprevalence data and indicator-based approaches" for consideration at PLOS Computational Biology.

As with all papers reviewed by the journal, your manuscript was reviewed by members of the editorial board and by several independent reviewers. In light of the reviews (below this email), we would like to invite the resubmission of a significantly-revised version that takes into account the reviewers' comments.

We cannot make any decision about publication until we have seen the revised manuscript and your response to the reviewers' comments. Your revised manuscript is also likely to be sent to reviewers for further evaluation.

Sincerely,

Benjamin Muir Althouse

Associate Editor

PLOS Computational Biology

Nina Fefferman

Deputy Editor

PLOS Computational Biology

Reviewer's Responses to Questions

**Comments to the Authors:**

Reviewer #1: The authors attempt to understand the variation in rubella’s basic reproduction number in countries for which pre-vaccination serological surveys are available in terms of measurable population characteristics. This is a thoughtful, carefully conducted, and important study with however disappointing results. Urban and rural differences would be expected by virtue of the relationship between contact rates and population density, and other patterns (e.g., education, healthcare, ...) probably are related. Other social phenomena should have greater impact than observed.

Some thoughts:

We know that the basic reproduction number is a property of a pathogen in a host population. One possible explanation for the as-yet-unexplained variation is that the pathogen is genetically heterogeneous (Rivailler et al. J Gen Virol 2017; 98:396-404). Another is that host age is far too influential to distinguish only children and adults, a possibility that the synthetic matrices of Prem et al. (PLoS Comput Biol 2017; 13:e1005697) make testable, assuming that mixing patterns haven’t changed. A third possibility is that the different assays (Dimech et al. Clin Microbiol Rev 2016; 29:163-74) that almost certainly were used obscured meaningful patterns.

Reviewer #2: Thank you for the opportunity to review “Estimates of the basic reproduction number for rubella using seroprevalence data and indicator-based approaches.” The authors conduct a detailed analysis estimating the basic reproduction number for rubella in 98 different settings using three different methodological approaches. The objectives were to estimate R0 for rubella and to investigate the effectiveness of using prediction methods via linear regression and machine learning regression to predict R0 as compared to a simple regional averaging approach. The motivation for this paper builds on the intrinsic value of knowing the basic reproduction number for any infectious disease in order to understand transmission dynamics and importantly inform control measures. They find that rubella R0 is generally low (<5) and that neither random forest regression nor linear regression out performed a simpler regional averaging approach. Compared to other published papers, R0 is a little lower in this analysis, but still within the range typically estimated for rubella. One significant insight is that for countries missing seroprevalence data, a regional averaging approach may provide a reliable estimate of R0.

The objectives are clearly laid out and addressed in the introduction and are original. The authors state that "the extent to which the regional average or particular indicators might predict what R0 might be for a given setting has not been studied for any infection." While other studies have evaluated associations between R0 and socio-demographic variables for other infectious disease, I am similarly unaware of any that evaluate the potential of particular indicators to predict R0 compared to other approaches. The methods are described in detail within the main text and supplemental information. The authors' use of multiple data sources that are publicly available is innovative and can be replicated for other (especially immunizing) infectious diseases. The methodology is rigorous and appropriate to answer the question at hand. Specially, the authors do a good job 1) including uncertainty in their estimates of R0 with bootstrapping and explaining within their limitations were uncertainly wasn't included and potential avenues for carrying over uncertainly in future work, and 2) exploring the impact of particular serological datasets on the correlation between indicators and R0. Lastly, the conclusions are consistent with the results presented.

I have a four minor revisions and three potentially major revisions.

Minor #1: Results first paragraph, description of Figure 1 findings. Please clarify where you are describing the country-specific R0 estimates, potentially specifying these as median estimates from the bootstrapped FoI estimates. It was just a little confusing to refer to the country-specific as point-estimates since the figure uses the term “point-estimate” for the regional estimates.

Minor #2: Results second paragraph. I think the first sentence needs to be revised to say that R0 was higher (rather than lower) for urban than for rural regions.

Minor #3: Results third paragraph, first sentence - Region of the Americas is listed twice.

Minor #4: Page 10, second paragraph within section on correlation b/w R0 calculated using country-specific seroprevalence data and indicators - the sentence reads, “When the simple linear regression was repeated by region, economic indicators were the most correlated with the basic reproduction number in Africa (Table 2).” Is it rather... when the simple correlation analyses were repeated by region ...?

Major #1: The broad findings from the work are based on the comparison of the prediction methods to the regional averaging (default) approach. I think the paper would benefit from a somewhat clearer understanding of the default approach. Specifically, what bootstrap-derived estimates of the force of infection are relied on in each analysis? In the non-imputed analysis described in S3 are the bootstrapped derived FoI estimates only from the settings in the three training folds, and does that explain why the MSE estimates vary across indicator in the non-imputed analysis compared to the imputed analysis? If true, are the FoI (from which to estimate a median R0) resampled 1000 times from the settings in the training folds, or are the FoI only a subset of the original 1000 bootstrapped samples? Also if true, how does this align with the discussion that states "The performance of this approach may have been overestimated given that bootstrap-derived estimates used to calculate the median would have included those from each of the studies being used to evaluate the performance," which reads as if bootstrap-derived estimates of the force of infection sampled from all 98 study sites were included in all default MSE estimates?

Major #2: On a similar note, if indeed bootstrap-derived estimates of the force of infection sampled from all 98 study sites were included in all default estimates, this does seem to be a relatively systematic bias that would decrease the MSE. Is it possible to conduct a sensitivity analysis for at least one indicator or subset of indicators assessing R0 MSE if you didn't include FoI estimates from the study site you are trying to predict?

Major #3: The authors state in the results that “For both assumptions about contact, estimates of R0 using the regional point estimate of the force of infection were usually smaller and led to a larger mean square error than the default R0 (Fig 1, Fig S2 and Table S5, Supplement),” but the significance of this finding is never revisited. It is possible to include a brief description of the motivation for this analysis and potentially the broad take away of this estimate under-performing compared to the regional averaging approach?

**Have the authors made all data and (if applicable) computational code underlying the findings in their manuscript fully available?**

Reviewer #1: Yes

Reviewer #2: **No: **The authors include references to data (that would need to be extrapolated), pseudocode, and in depth details of their methods, but do not actually include the data and code that can be used to easily replicate their findings.

PLOS authors have the option to publish the peer review history of their article (what does this mean?). If published, this will include your full peer review and any attached files.

Reviewer #1: No

Reviewer #2: No
---

## [Decision Letter · Decision Letter 1]

23 Jan 2022

Dear Dr Papadopoulos,

We are pleased to inform you that your manuscript 'Estimates of the basic reproduction number for rubella using seroprevalence data and indicator-based approaches' has been provisionally accepted for publication in PLOS Computational Biology.

Best regards,

Benjamin Althouse

Associate Editor

PLOS Computational Biology

Nina Fefferman

Deputy Editor

PLOS Computational Biology

Reviewer's Responses to Questions

**Comments to the Authors:**

Reviewer #1: I have no additional comments.

Reviewer #2: Thank you kindly for addressing my comments thoroughly. It is a very nice paper and I appreciate the opportunity to review it.

**Have the authors made all data and (if applicable) computational code underlying the findings in their manuscript fully available?**

Reviewer #1: Yes

Reviewer #2: Yes

PLOS authors have the option to publish the peer review history of their article (what does this mean?). If published, this will include your full peer review and any attached files.

Reviewer #1: No

Reviewer #2: No

---

## [Editor Report · Acceptance letter]

9 Feb 2022

PCOMPBIOL-D-21-00482R1 

Estimates of the basic reproduction number for rubella using seroprevalence data and indicator-based approaches

Dear Dr Papadopoulos,

I am pleased to inform you that your manuscript has been formally accepted for publication in PLOS Computational Biology. Your manuscript is now with our production department and you will be notified of the publication date in due course.

With kind regards,

Katalin Szabo
